# Contribution of retrotransposition to developmental disorders

Eugene J. Gardner [1], Elena Prigmore[1], Giuseppe Gallone[1], Petr Danecek[1], Kaitlin E. Samocha[1], Juliet Handsaker[1], Sebastian S. Gerety [1], Holly Ironfield[1], Patrick J. Short [1], Alejandro Sifrim[2], Tarjinder Singh [1], Kate E. Chandler[3], Emma Clement[4], Katherine L. Lachlan[5,6], Katrina Prescott[7], Elisabeth Rosser[4], David R. FitzPatrick[8], Helen V. Firth[1,9] & Matthew E. Hurles[1]*

Mobile genetic Elements (MEs) are segments of DNA which can copy themselves and other transcribed sequences through the process of retrotransposition (RT). In humans several disorders have been attributed to RT, but the role of RT in severe developmental disorders (DD) has not yet been explored. Here we identify RT-derived events in 9738 exome sequenced trios with DD-affected probands. We ascertain 9 de novo MEs, 4 of which are likely causative of the patient's symptoms (0.04%), as well as 2 de novo gene retro-duplications. Beyond identifying likely diagnostic RT events, we estimate genome-wide germline ME mutation rate and selective constraint and demonstrate that coding RT events have signatures of purifying selection equivalent to those of truncating mutations. Overall, our analysis represents a comprehensive interrogation of the impact of retrotransposition on protein coding genes and a framework for future evolutionary and disease studies.

[1] Wellcome Sanger Institute, Wellcome Genome Campus, Cambridge, Hinxton CB10 1SA, UK. [2] Department of Human Genetics, KU Leuven, Herestraat 49 Box 602 Leuven B-3000, Belgium. [3] Manchester Centre for Genomic Medicine, Manchester University Hospitals NHS Foundation Trust, Manchester Academic Health Science Centre, Manchester, Greater, Manchester M13 9WL, UK. [4] Department of Clinical Genetics, North East Thames Regional Genetics Service, Great Ormond Street Hospital for Children NHS Trust, Holborn, London WC1N 3JH, UK. [5] Wessex Clinical Genetics Service, Southampton University Hospitals NHS Foundation Trust, Princess Anne Hospital, Southampton SO16 5YA, UK. [6] Faculty of Medicine, Human Development and Health, University of Southampton, Southampton SO17 1BJ, UK. [7] Clinical Genetics Department, Yorkshire Regional Genetics Service, Leeds Teaching Hospitals NHS Trust, Chapel Allerton Hospital, Leeds LS7 4SA, UK. [8] MRC Human Genetics Unit, MRC IGMM, University of Edinburgh, WGH, Edinburgh EH4 2SP, UK. [9] East Anglian Medical Genetics Service, Box 134, Cambridge University Hospitals NHS Foundation Trust, Cambridge Biomedical Campus, Cambridge CB2 0QQ, UK. *email: meh@sanger.ac.uk

In humans, three classes of mobile genetic elements (MEs)—*Alu*, long interspersed nuclear element 1 (L1), and SINE-VNTR-*Alu* (SVA)—are still active and can generate new copies, known as mobile element insertions (MEIs), throughout their host genome[1]. The L1 replicative machinery can also facilitate the duplication of non-ME transcripts, typically protein-coding genes, through the mechanism of retroduplication to generate processed pseudogenes (PPGs)[2]. Combined, these two processes constitute retrotransposition (RT) in the human genome, with new (de novo) MEI variants previously estimated to occur in every 1 out of 18.4–26.0 births[3]. On a population level, each individual human genome harbors ~1200 polymorphic variants, with the smallest ME, *Alu*, generally contributing 75% of total RT polymorphisms[4–6].

To date, roughly 130 pathogenic variants caused by RT activity have been documented[7]; however, the majority of these deleterious events have been discovered in isolated cases. Neither MEIs nor PPGs are canalyzed as part of routine clinical sequencing, and thus represent a largely unassessed category of genetic variation in many disorders. Furthermore, of the clinically relevant RT-attributable cases thus identified, few (~14/123; 11.4%) are caused by new mutational events and are instead typically attributable to rare inherited polymorphisms[7]. In addition, of the large disease-focused whole-genome sequencing (WGS) projects which have ascertained MEIs, all have focused on autism[8–10] and have failed to identify likely causative RT-derived variants. In fact, in the largest and most recent WGS study investigating the role of large structural variants in the genetic architecture of autism, the authors failed to identify a single de novo MEI in a coding exon, deleterious or otherwise, in 829 families[9]. This finding is likely a result of several factors, predominant among them the low frequency of cases attributable to gene disruption by MEIs in autism[10], due in part to a low ME mutation rate[3] and lack of a sufficiently large sample size[8,9,11]. As such, it is not precisely known at what rate de novo ME variants are generated in the human genome, the functional consequences of such variants, the role that they play in the etiology of rare disease, and if routine clinical sequencing should assess patient genomes for deleterious RT events.

Herein, we analyze the whole-exome sequencing (WES) data produced by the Deciphering Developmental Disorders (DDD) study to systematically assess the role of RT in severe developmental disorders (DDs). The DDD data have already been investigated for pathogenic single-nucleotide variants (SNVs), small insertions and deletions (InDels), large copy number variants (CNVs), and other classes of variation[12–18]. Approximately, 24% of DDD cases harbor a pathogenic de novo mutation in a gene known to be associated with developmental disorders[12]. The DDD cohort should thus be relatively enriched for highly penetrant de novo RT events in comparison with recent studies on autism[8,9]. With a cohort of 9738 trios ($n = 28,132$ individuals) whole-exome sequenced, the DDD study presents a powerful opportunity to identify, and ascertain the role in DD of, pathogenic de novo RT events that impact coding sequences.

## Results

**Generation of a genome-wide data set of RT variants.** To assess 9738 DDD study trios for RT events, we utilized two separate computational approaches to identify both MEIs and PPGs. First, we used the mobile element locator tool (MELT)[5] to identify *Alu*, L1, and SVA variants located within the WES bait regions (see the Methods section). The second is a new bespoke tool developed to identify PPGs from the WES data (Methods, Supplementary Fig. 1). Due to cross-hybridization between a PPG and the exome baits targeting the donor gene, we anticipated that we should be able to detect PPGs genome wide, not just the subset that insert within the WES bait regions. Our PPG detection tool ascertained putative PPGs by identifying multiple discordant read pairs mapping to different exons of the same transcript, before then typing all individuals for the presence/absence of the PPG using discordant read pairs and split reads. The tool was optimized by comparing against previously described PPG polymorphisms in the 1000 genomes project (1KGP; see below).

As our study is the first to discover MEIs directly from WES on a large scale with MELT, we first utilized matched sample WGS data to determine if MELT could ascertain MEI variants reliably from the WES data. We compared MEI variants identified by MELT within the DDD WES data to both WGS data generated for 90 overlapping DDD individuals and population MEI data previously generated from the 1000 Genomes Project Phase 3 (1KPG)[4–6] WGS data. The latter comparison was to ensure that the number of exonic MEIs identified within DDD WES data was concordant with expectations at the individual and population level. When comparing our WES genotypes to WGS in the same individuals, we had a genotype concordance rate of 94.46% (93.93% *Alu*, 97.29% L1, 98.25% SVA) among calls with at least 10× coverage in our WES data. In total, we were able to re-identify 1450 (1289 *Alu*, 160 L1, 1 SVA) MEI genotypes, or 84.5% of all heterozygous or homozygous genotypes identifiable with WGS in WES bait regions (Supplementary Table 1). Based on these findings, we were confident that MELT was appropriately calibrated to ascertain MEIs in WES data.

We identified 1129 MEI variants and 576 polymorphic PPGs, with each individual's exome containing on average 33.0 ± 5.0 (SD) variants (Table 1). All MEIs were genotyped across all individuals to form a comprehensive catalogue of RT-derived variation within and adjacent to (±50 bp) sequences targeted in the WES assay (Methods), including coding exons and targeted noncoding elements (Table 1; Fig. 1). The average time to assess a single family for RT-derived events was ~15 min, and the rate of false findings was low (1 incorrect candidate de novo variant per every 295 patients; either a false-positive variant [1 per 649] or false-negative genotype [1 per 541] in at least one parent). We investigated whether false-negative parental genotypes could be

**Table 1 RT variant discovery in the DDD study**

|  | Total sites cohort-wide | Mean sites per unaffected parent | Total de novo sites |
|---|---|---|---|
| Alu | 917 | 23.6 ± 4.2 | 7 |
| LINE-1 | 167 | 2.8 ± 1.5 | 2 |
| SVA | 45 | 0.2 ± 0.5 | 0 |
| *Total – MEI* | *1129* | *26.6 ± 4.7* | *9* |
| Processed pseudogenes (PPGs) | 576 | 6.6 ± 1.7 | 2 |
| *Total – MEI + PPG* | *1705* | *33.0 ± 5.0* | *11* |

Quantification of the four different classes of retrotransposons discovered as part of this study. Rows in italic indicate totals across the classes listed above

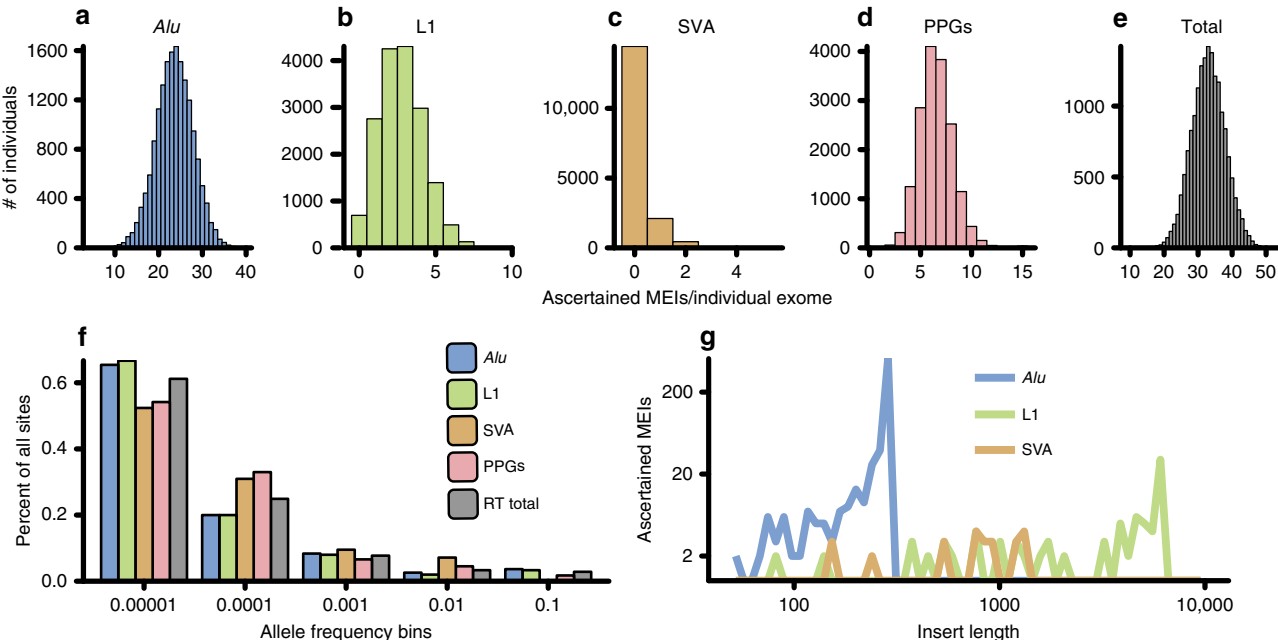

**Fig. 1** The DDD RT call set. **a–e** Histograms of the total number of variants per individual for the four classes of RT events identified in the DDD cohort (Alu: blue; L1: green; SVA: orange; PPGs: red; combined RT events: grey) in size one bins. **f** Allele frequency distributions for the RT classes depicted in **a–e** in $\log_{10}$ allele frequency bins. **g** Insert size estimates provided by MELT for the MEI classes ascertained in this study in $\log_{10}$ insert size bins. All plots only include variants from unaffected parents

mosaic. Using long-PCR coupled to long-read sequencing (see methods), we were able to phase seven of our false-negative Alu variants to nearby SNVs. With our approach, we determined that one was part of a complex event of unknown mosaicism (FCGBP: c.4743_4744insAlu(1_110)), and six were constitutive in both parent and proband (Supplementary Fig. 2; Supplementary Data 2).

As expected, the total number of variants per individual for each RT class (Fig. 1a–d) as well as combined number of RT events (Fig. 1e) approximated a Poisson distribution (see the Methods section; Supplementary Fig. 3). The vast majority of variants are rare (allele frequency $< 1 \times 10^{-4}$; Fig. 1f), with $> 65\%$ of Alu and L1 variants identified in fewer than four unrelated individuals. SVA and PPGs appear to be moderately under ascertained compared with Alu and L1 at lower AFs, with $> 50\%$ of variants identified in the lowest AF bin and length estimates for the three MEI classes largely fit the findings of previous studies (Fig. 1g)[4,5].

We next sought to ensure that our total number of ascertained RT variants, both on a population and individual basis, accorded with previously published WGS data[4,5]. On a population level, WES did not appreciably limit our overall sensitivity compared with the WGS sampled data. When we compared a downsampled version of our call set to the 1KGP, our total number of Alu and SVA variants fell within the expected distribution, while L1 was close to expectation (Supplementary Fig. 4).

To assess the quality of the PPG call set, we compared PPG allele counts (i.e., total number of individuals with a retroduplication of a given gene) to a recent assessment of PPGs in samples sequenced as part of the 1KGP[6]. Generally, PPGs identified in both data sets shared similar relative allele counts ($r^2 = 0.64$) and variants identified in this study, but missing from Zhang et al.[6] are typically rare (Supplementary Fig. 5). To further validate our approach and ensure that the identified PPG donor genes fit with previously identified patterns of germline PPG formation[2,19], we assessed each donor gene for both functional annotation and expression across 30 tissue types analyzed by the GTEx consortium[20]. The major functional cluster (DAVID[21] enrichment score 8.82) belonged to

genes involved in the ribosomal and translational machinery, consistent with previous findings involving fixed PPGs in the human genome[2]. Our expression analysis likewise confirmed previous findings[19], and shows that donor genes that give rise to PPGs are more highly expressed in a large number of tissues compared with non-retroposed genes (Wilcoxon rank sum $p < 1 \times 10^{-3}$ for all tissues; Supplementary Fig. 6). In addition, while it could be assumed that increased germ-line expression of a gene may play a role in increased probability of PPG generation, when we compared PPG donor gene expression in the testis and ovary to that in other tissues, the majority of tissues (19/28, identical tissues for ovary and testis) showed statistically identical patterns of donor gene expression (Wilcoxon rank sum $p > 1 \times 10^{-3}$; Supplementary Fig. 6).

**Coding RT burden and constraint.** As expected for WES, the vast majority (84.9%) of detected MEIs impacted the coding or intronic sequence of a protein-coding gene or a regulatory element targeted in the augmented WES assay described in Short et al.[15] (Fig. 2a). Due to the large numbers of individuals with WES data in this study, we have ascertained over five-fold more exonic variants than the largest previously published study (Supplementary Fig. 7)[4,5]. Nonetheless, the number of MEIs identified in this study, based on the proportion of the genome assayed, likely represent only 2.2% of MEI variants genome wide in these individuals.

Our large collection of coding variants allowed us to examine the evolutionary forces acting on coding MEI variation (Fig. 2b). To examine selective constraint, we utilized two common measures: the proportion of variants observed in only one individual (e.g., singletons)[22] and the proportion of variants found in genes likely to be intolerant of loss of function (LoF), as determined by the pLI score[23]. To avoid issues of relatedness and the potential for clinical ascertainment bias for pathogenic MEIs in individual DD patients, only the 17,032 unaffected parents sequenced as part of DDD were included in our analysis. MEIs which directly impact exons are under strong selective constraint,

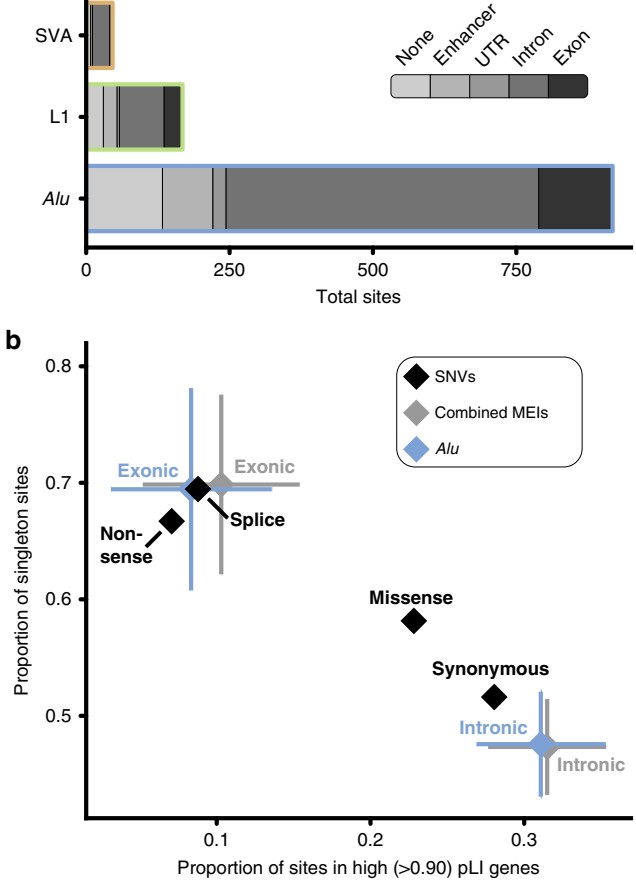

**Fig. 2** Coding constraint on MEIs. **a** Cumulative consequence annotations for *Alu*, L1, and SVA MEIs in all samples ($n = 28,132$ individuals) analyzed. The majority of variants identified in this study fell within the noncoding space (either an enhancer or intron). **b** Comparison of constraint between MEIs and SNVs in unaffected parents. To compare the impact of exonic and intronic *Alu* (blue) and all MEIs (grey) to varying classes of SNVs (black), we used two metrics: the proportion of variants in genes that have been identified as LoF intolerant as gauged by pLI-score[22] (x-axis) and the proportion of variants identified in only one individual (i.e., singletons; y-axis). Error bars indicate 95% confidence intervals based on population proportion; confidence intervals were calculated for SNVs, but are too small to appear at the resolution displayed in this figure

indistinguishable from that of both nonsense and essential splice site SNVs (Fig. 2b). Interestingly, we did not find any sign of selection acting on intronic MEIs as they appear to be constrained similarly to synonymous SNVs. In contrast to previous studies[24,25], we did not find a statistically significant ($\chi^2$ $p < 0.05$) bias toward intronic MEIs inserted in the antisense orientation of the gene in which they are found (Supplementary Fig. 8). This is likely not a repudiation of such work, but attributable to the relatively small number of intronic events we identified as part of our analysis compared with WGS[4,24] or reference genome-based[25] studies. To put our findings on exonic MEI constraint into perspective with other forms of variation, every human genome will harbor approximately one ($0.76 \pm 0.62$ SD per individual) MEI which directly impacts protein-coding sequence. Since MEIs are similar to nonsense SNVs in terms of deleteriousness (Fig. 2b), MEIs thus make up 0.6%[22,26] of all coding protein-truncating variants (PTVs; among SNVs, InDels, and large CNVs) in each individual human genome.

While we were unable to perform similar population genetic analyses for PPG events, due to the difficulty of resolving the putative insertion site with WES data and thus distinguishing between different PPGs for the same donor gene, we were able to assess the propensity for specific genes to give rise to PPGs based on their selective constraint. We observed that PPG donor genes were significantly enriched for genes that are highly intolerant of loss of function variation (pLI > 0.9). High pLI genes make up 25.3% of donor genes, compared with 17.6% of all protein-coding genes (Fisher's $p = 2.6 \times 10^{-4}$)[22]. This observation is likely driven by loss-of-function intolerant genes being more likely to be highly expressed in multiple tissues[22], similar to genes known to have been retroduplicated (Supplementary Fig. 6)[19]. This observation implies that PPG events rarely strongly perturb the function of their donor gene—despite several previously documented instances of PPGs impacting expression or functionality of their donor gene[27].

**Clinical annotation of de novo RT variants in DD.** Using the computational approaches outlined above, we identified a total of 11 germ-line de novo RT variants (Table 2). Our findings include coding, noncoding, pathogenic, and benign variants, as well as, to our knowledge, the first de novo MEI identified in a pair of monozygotic twins (Supplementary Fig. 9). All de novo RT variants were confirmed via a PCR assay specific to the RT class (Fig. 3; Supplementary Fig. 9; Supplementary Fig. 12) and, where possible, inspected for poly(A) tail and target site duplication—hallmarks of bona fide RT activity[28]. We identified no de novo RT variants which localized to the noncoding elements included on the WES capture, which falls in line with expectations based on mutation rate estimates (Fig. 4b). We also attempted to determine the parental origin of each RT event using SNVs located on sequencing reads which support the RT insertion (Table 2). Of the 11 de novo RT events, we were able to phase three variants, all to the father. While this finding is not statistically significant ($\chi^2$ $p = 0.083$), it fits with previous findings that the majority of de novo structural variants[9], and indeed most variant classes[29], are attributable to paternal origin.

Nine of our validated de novo mutations were MEIs (7 *Alu* and 2 L1), or a rate of approximately one de novo event per every 1000 patient exomes sequenced (9/9738). As expected, based on both the total number of polymorphisms[3–5] and mutation rate (Table 1; Supplementary Table 3), we identified more *Alu* de novo variants than the other RT classes. We also identified two PPG germ-line de novo variants, or approximately one new PPG per every 5000 patient WGS (2/9738). As a further quality control for PPGs, we capillary sequenced all resulting PCR products to confirm the gene of origin (Supplementary Fig. 12) and performed WGS to identify the PPG insertion site. We were able to localize the *SERINC5* PPG to an ~50 Kbp intron of the gene *CLIC4* and the *SLC35F2* event to an intergenic region between the genes *MAK* and *GCM2* (Fig. 3e). Neither of the events directly impacted coding sequence and *CLIC4* is neither under strong selective constraint nor known to have any link with DD.

Each de novo mutation was then compared with known DD-associated genes (using the Developmental Disorders Genotype-to-Phenotype database–DDG2P) to identify potentially pathogenic variants (Table 2). Of the mutations identified, four directly inserted into coding exons of DD-associated and fetal brain-expressed (see the Methods section; Supplementary Table 4)[30] genes (Fig. 3, Table 2) with all four found in genes statistically enriched for PTVs[12] and therefore likely to operate by a LoF mechanism. We did not identify any intronic de novo mutations likely to be pathogenic (Fig. 3a–d; Supplementary Fig. 9). An additional mutation inserted into the coding sequence of a

**Table 2 Confirmed germ-line de novo variants in the DDD study**

| Insertion coord. | RT type | Genomic compartment | ENSEMBL gene ID | HGNC gene ID | pLI | DDG2P annotation | Decipher ID[57] | Diagnostic? | Parental origin | Notes |
|---|---|---|---|---|---|---|---|---|---|---|
| chr3:9495459 | Alu | Exonic | ENSG00000168137 | SETD5 | 1.000 | Confirmed, monoallelic | 280818 | True | Father | |
| chr5:176638159 | Alu | Exonic | ENSG00000165671 | NSD1 | 1.000 | Confirmed, monoallelic | 259118 | True | Unknown | Included in Wright et. al.[58] |
| chr6:159190834 | Alu | Exonic | ENSG00000092820 | EZR | 0.988 | None | 300984 | False | Unknown | |
| chr7:77552086 | Alu | Exonic | ENSG00000006576 | PHTF2 | 0.024 | None | 271388 | False | Father | |
| chr3:135913800 | Alu | Intronic | ENSG00000174579 | MSL2 | 0.890 | None | 292325 | False | Unknown | |
| chr3:148614204 | Alu | Intronic | ENSG00000163751 | CPA3 | <0.001 | None | 270426; 270428 | False | Unknown | Monozygotic twins |
| chr3:172480619 | Alu | Intronic | ENSG00000114346 | ECT2 | <0.001 | None | 307591 | False | Unknown | |
| chr12:46246325 | L1 | Exonic | ENSG00000189079 | ARID2 | 1.000 | Probable, monoallelic | 264759 | True | Unknown | |
| chr5:88100580 | L1 | Exonic | ENSG00000081189 | MEF2C | 0.004 | Confirmed, monoallelic | 285645 | False | Unknown | |
| chr6:10847968 | Retrogene-SLC35F2 | Intergenic | N/A | N/A | N/A | N/A | 291670 | False | Unknown | |
| chr1:25074202 | Retrogene-SERINC5 | Intronic | ENSG00000169504 | CLIC4 | 0.009 | None | 301168 | False | Father | |

Relevant clinical and annotation information for MEI and PPG de novo variants identified as part of this study. Location of the insertion event is given in human build GRCh37 reference coordinates (Insertion coord.). A true value in the Diagnostic column indicates, at the time of publication, that this variant intersected a known DD gene, and was deemed likely to be involved in the patient's phenotype by the referring clinician; false does not indicate whether or not, with additional future evidence, the gene may become associated with DD and the variant thus deemed diagnostically relevant. If applicable, ENSEMBL[59] gene IDs indicate the gene impacted, not the gene from which the event is derived (i.e., for PPGs)

strongly LoF-intolerant gene, *EZR* (pLI = 0.99; Supplementary Fig. 9), but we could not directly attribute it to the patient's phenotype due to lack of significant enrichment for PTVs, although there is prior evidence for a role in a familial DD syndrome[31]. The four mutations in DD-associated genes were reported to the referring clinician for clinical interpretation based on both initially reported and updated phenotypes (Supplementary Table 2). Three out of four reported mutations (*NSD1*, *MEF2C*, and *ARID2*) were subsequently deemed to be likely causative of the patient's phenotype (Supplementary Table 2) by the referring clinician. To determine if our patient with a likely pathogenic *NSD1* variant phenocopies the already described aberrant blood DNA methylation signature typical of Sotos syndrome patients[32], we compared our patient's genome-wide methylation status to several already published Sotos syndrome cases (see the Methods section)[32]. Our patient strongly clusters with other confirmed Sotos syndrome patients and confirms our clinical assessment and diagnosis of this particular patient (Supplementary Fig. 10). The fourth patient, with an *Alu* insertion in *SETD5* (Fig. 3a), has clinical features (polydactyly and truncal obesity; Supplementary Table 2) more suggestive of a ciliopathy. As such, the identified MEI is unlikely to be the sole cause for the patient's DD, but may contribute to a composite phenotype. Accordingly, we did investigate this patient's exome for other potentially diagnostic variants, but did not identify additional de novo or inherited variation that could plausibly be associated with this patient's phenotype.

We also examined our data set for inherited rare pathogenic RT variants. We evaluated variants inherited from an affected parent, bi-allelic inheritance (either a homozygous MEI or a heterozygous MEI paired with another variant class), and X-linked variants maternally inherited by affected males. We did neither identify any rare MEI variants inherited from an affected parent nor any compound heterozygous individuals with a rare MEI and a non-MEI PTV (e.g., SNV/InDel) impacting the same gene. We did identify a single proband-specific homozygous MEI inserted into an exon of *PAN2* which was unique to a single family. This gene was recently identified as nominally significant (genome wide $p = 4.2 \times 10^{-4}$) in a study investigating the role of recessive variants in DD[13], although more data are required to be confident of its association to DD. We also identified a total of 22 (14 *Alu*, 7 L1, and 1 SVA) polymorphic MEIs on the X chromosome, of which 4 (3 *Alu* and 1 L1) directly impacted protein-coding sequence. Of these variants, none were at a low enough allele frequency to be reasonably DD associated, were located within a gene associated with DD, nor fit an inheritance pattern consistent with X-linked disease.

**Mutation rate and enrichment of deleterious RT events in DDD.** Based on our findings, in the coding and peri-coding portion of the genome, one out of every 2434 DD cases (0.04% ± 0.04; 95% CI) is directly attributable to RT-derived mutagenesis. To determine both if our observed number of de novo variants meets expectation and if our patient cohort is enriched for causal de novo RT events, we estimated the population mutation parameter, $\Theta$[33], from the unaffected parents in the DDD study and from the 1KGP[4,5] (Supplementary Table 3). The resulting calculation gives very similar estimates of MEI mutation rate (combined across *Alu*, L1, SVA) of between $1.4 \times 10^{-11}$ (1KGP) and $1.2 \times 10^{-11}$ (DDD) variants per bp per generation ($\mu$), or ~1 new MEI genome-wide per every 12–14 births—largely concordant with prior estimates from WGS data sets[3,34,35].

Using this genome-wide mutation rate, we estimated the number of expected mutations in various genomic compartments, including within genes intolerant to PTVs and within

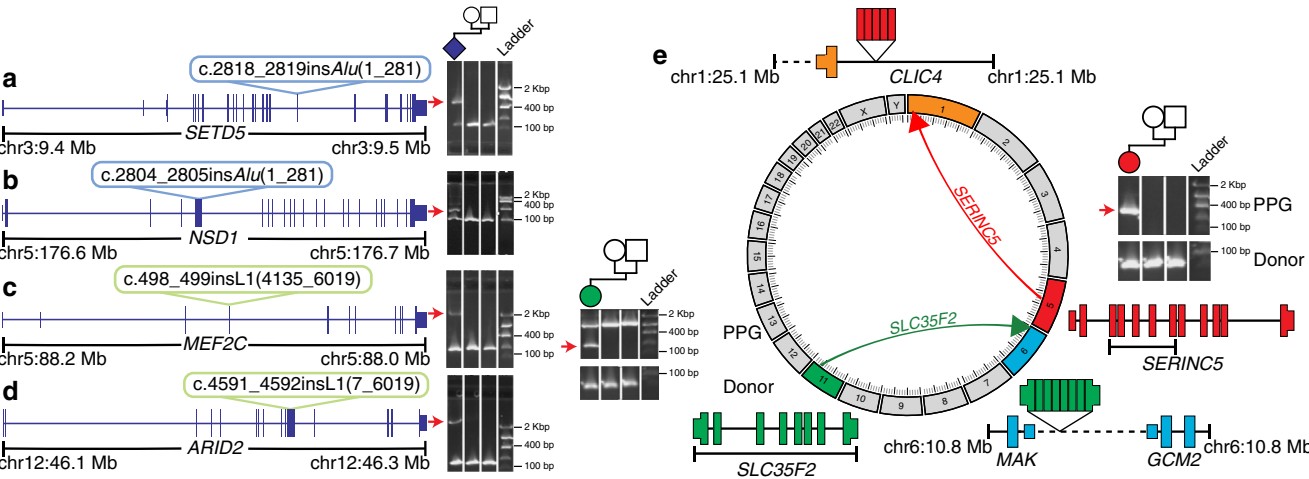

**Fig. 3** RT-derived de novos in the DDD. We identified a total of nine de novo MEIs, four of which disrupted the protein-coding sequence of a known DD gene: **a** *SETD5*, **b** *NSD1*, **c** *MEF2C*, and **d** *ARID2*. Shown in each panel is a diagram of the affected gene (blue model) with the relevant insertion indicated with a colored bubble. To the right are PCR validations confirming the de novo status of each mutation; a positive result is indicated by a raised secondary band present only in the proband sample (red arrow). **e** Circos diagram and PCR results for two identified germ-line de novo PPGs. For each de novo PPG shown is a diagram of the donor gene (gene model), location of duplication as PPG (directional arrow), and new insertion site. Exons from the donor gene included in the PPG are indicated by brackets underneath the donor gene model. To confirm PPG presence, PCR was performed (Methods) on proband, paternal, and maternal gDNA (sample in each lane is shown by pedigree). The band which represents the PPG is marked with a red arrow, and was confirmed via capillary sequencing (Supplementary Fig. 12). Dashed lines indicate intergenic regions, all genes models are shown in sense orientation, and PPG gene diagrams are not to scale

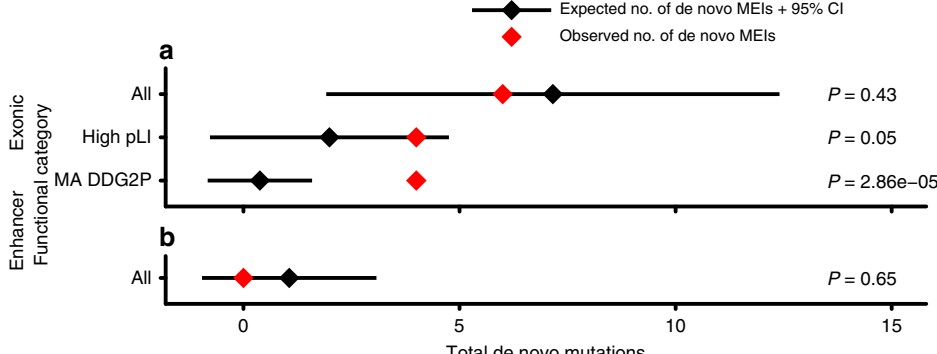

**Fig. 4** Estimating enrichment of deleterious MEIs. Depicted are the total number of expected (black) and observed (red) de novo mutations observed in exons (**a**) and enhancers (**b**) for all, high pLI (pLI > 0.9), and known monoallelic DD (MA DDG2P) genes. Expectation is based on the Poisson distribution of 100 simulations utilizing the neutral mutation rate ($1.2 \times 10^{-11}$ $\mu$). *P*-values are based on the Poisson distribution and used to determine statistical deviation of observed to expected de novo counts for exons and enhancers

DD-associated genes (Fig. 4; Methods). We identified a significant enrichment of de novo MEIs in dominant DD-associated genes (Poisson $p = 2.86 \times 10^{-5}$), but not in the much larger set of LoF intolerant genes (Poisson $p = 0.05$). To ensure that this finding was not due to inaccurate estimation of the genome-wide mutation rate, we also assessed the probability that four out of six exonic de novo MEIs would fall within exons of dominant DD-associated genes by chance, based on the proportion of the exome represented by these genes (and assuming known DD-associated genes have the same MEI mutation rate as other genes) and likewise found a significant enrichment ($p = 4.3 \times 10^{-5}$; see the Methods section).

## Discussion

Here, we have described the development, validation, and exemplification at scale of an analytical pipeline for the rapid assessment of patient genomes for RT variants. We have used these approaches to present the largest study examining the

coding genome for RT-derived variation to date (Table 1; Fig. 1). With this data set, we first demonstrated that exonic MEIs (regardless of insertion length) are under selective constraint on par with protein-truncating SNVs (Fig. 2). We identified four likely pathogenic RT mutations, two *Alu* and two L1 insertions (Fig. 3), all of which arose de novo in known haploinsufficient DD-associated genes (Fig. 3a–d), implying that dominant loss-of-function is the major mode of pathogenic exonic RT variation. Finally, we estimated the genome-wide MEI mutation rate and used it to determine that DDD probands are enriched for damaging RT variation within exons of dominant DD-associated genes (Fig. 4a).

The total number of polymorphic, exonic RT variants identified in DDD is concordant with previous studies characterizing MEI variation[3,5,36]. Pathogenic MEIs make up 0.04% of diagnoses in the DDD study (4/9,738 probands), a small yet individually significant collection of diagnostic variants. We additionally investigated our data for causative, inherited MEIs, and did not identify any such events. We infer that despite making up a

significant proportion of reported MEI variants in the clinical literature[7], bi-allelic or X-linked MEI events are a less frequent class of pathogenic variant in developmental disorders. This is in keeping with recent estimates[13] that in a largely outbred clinical population, such as in the UK, recessive disorders caused by coding variants account for a much smaller fraction of patients than dominant disorders. We also attempted to investigate both the parental origin of de novo variants (Table 2) and the mosaicism of MEIs with low variant allele frequency (Supplementary Fig. 2). In both cases, additional work incorporating long-read technology is needed to ascertain parental contribution and, in the case of parental mosaic MEIs, potential recurrence risk.

Interestingly, it appears that the contribution of diagnostic RT variants may vary among diseases. Wimmer et al.[37] reported a total of 13 diagnostic, exonic MEI variants in 4500 neurofibromatosis type I patients (0.3% of patients). This rate is seven times higher than that observed in DDD, and was attributed to a potential RT mutation hotspot associated with the canonical L1 endonuclease cleavage site of 3′-AA/TTTT-5′[38] within the neurofibromatosis-associated gene, NF1. A recent study using a cell culture retrotransposition assay to study MEI site preference was unable to identify a mutation hotspot in NF1[39], and therefore suggests further work is needed to investigate the role of sequence context in determining the overall genomic landscape of RT-mediated disease. Analogously, inclusion of sequence context into the SNV mutation model noticeably improved the ability to determine enrichment/depletion of deleterious SNVs within genes[22,23].

Our study is limited in that by analyzing exome sequences we only identified ~2% of the RT variants in each individual human genome studied[4,5]. Despite a number of known disease-associated intronic MEIs in the literature, we did not identify a pathogenic intronic MEI. As such, it remains an open question as to what contribution RT mutations in the noncoding genome play in the etiology of DD. While it appears that the contribution of regulatory elements to DD is relatively small, as defined by this (Fig. 4b) and other studies[15], previous work has identified a significant signature of purifying selection against MEI events within 100 bps of exons[25]—variants which our study could potentially identify. As our data suggest that the majority of DD cases with pathogenic coding MEIs are due to de novo insertions (Table 2; Fig. 3), we conjecture that most additional DD-associated MEIs may be located in the introns of known DD-causing genes and disrupt splicing—a known disease mechanism attributable to RT-derived mutagenesis[7,37,40]. Simulations suggest that under a null genome-wide mutation model, we should expect to observe 12.3 (5.4–19.2, 95% CI) de novo intronic RT mutations in dominant DD-associated genes in a population sample of 9738 individuals. As such, a WGS study of a clinical population of similar size to that analyzed here should be well powered to estimate the pathogenic contribution of intronic MEIs.

De novo MEIs are typically readily interpretable with modest informatics expertise, and represent a clinically relevant class of variation to assay in clinical bioinformatics pipelines. While we ultimately find that the overall burden of RT-attributable disease is relatively low in the human population, it is nonetheless an important consideration when elucidating the genetic basis of disease in individual patients.

## Methods
**Patient recruitment and sequencing.** A total of 13,462 patients were recruited from 24 clinical genetics centers from throughout the UK and the Republic of Ireland as previously described[41]. Informed consent was obtained for all families, and the study was approved by the UK Research Ethics Committee (see the Acknowledgements section). For the purposes of this study, individuals that were not recruited as part of a trio (e.g., individual patients or patients with just one parent), were included on the DDD sample blacklist, or failed to meet MELT QC requirements[5] were excluded from downstream analysis (leaving $n = 9738$ probands; 28,132 individuals). Sequencing of families and alignment to build GRCh37 of the human reference genome with bwa[42] were performed as previously described[12].

**Processed pseudogene pipeline development.** PPGs, particularly young polymorphic events, share highly homologous sequence with the source gene from which they are derived. Consequently, the WES bait capture method will capture both DNA from the original donor gene and the new daughter copy. This allows, compared with our approach for MEI discovery, for ascertainment of PPGs genome wide. While this approach does come with limitations, such as difficulty in identifying insertion variants, we can still determine events per individual.

Our discovery pipeline functions in two steps: first we collect read evidence on an individual level to determine which genes have been retroduplicated in that individual (Supplementary Fig. 1). Second, we determine presence/absence of each PPG in every individual in the DDD cohort based on the gene models built in the first step. In step one, we iterate over all genes in the ENSEMBL gene database which have a determined pLI score[22] and collect discordant read pairs (DRPs) which map between exons and have an insert size > 99.5% of all other reads in the sample. If more than four reads linking two exons are found, the gene is considered to be retroduplicated elsewhere in the genome. In step two, for each gene identified in step one, all evidence across all PPG-positive individuals are pooled to make a model of the PPG. This model is then used to check for DRP and split read pair (SRP) evidence in all genomes. If an individual has at least five total read pairs of supporting evidence with at least one SRP and one DRP, an individual is considered positive for the given PPG. All genes and individuals were combined into a flat file listing presence or absence of a given PPG in each individual.

**MEI call set generation and consequence annotation.** To identify MEIs in the DDD WES data, we utilized the previously published MELT[5]. MELT was run with default parameters (except the "-exome" flag during IndivAnalysis) using "Split" mode to generate a final unified VCF-format file[43] of all 28,132 unfiltered individuals independently for each MEI type (Alu, L1, SVA). Following initial data set generation, we found that a subset of variants internal or adjacent to (±50 bp) low complexity repeats (defined here as a run of sequence ≥15 bp composed of two or fewer nucleotides) were likely false positive. As such, we added an additional filter to the final MELT VCF, lc (low complexity), which removes such false positives from downstream analysis. Variants that could not be genotyped in at least 25% of individuals, had ≤ 2 split reads, had MELT ASSESS score < 3, or had any value in the VCF FILTER column other than PASS or rSD were filtered.

To determine if variants per individual approximated a Poisson distribution, we calculated the mean number of sites per unrelated individual ($\lambda$) independently for each RT class and for all RT classes combined as displayed in Fig. 1a–e. Our calculated $\lambda$ was then supplied to the function rpois in R to generate a random deviate and plotted alongside histograms identical to Fig. 1a–e in Supplementary Fig. 3.

To generate consequences plotted in Fig. 2a, all MEIs were annotated using Variant Effect Predictor v88 (VEP)[44] on human genome build GRCh37 and intersected with bedtools intersect[45] to enhancers (one of heart[46], VISTA[47], or highly evolutionarily conserved[48]) included on the DDD WES capture[15]. Only a single consequence was retained for each variant, with priority given to enhancer annotation. Primary transcript as determined by VEP was used for all gene-based consequences, pLI score[22] annotation, and DDG2P disease association (Table 2).

**Assessment of potentially mosaic MEIs.** To determine the mosaicism of variants that were identified as false-negative de novo during initial MELT analysis (i.e., genotyped as heterozygous in a single proband and homozygous reference in both parents), we randomly selected 1000 trios and determined the total number of reads supporting both the insertion and reference allele (i.e., allelic proportion) at all 917 Alu loci identified in this study. Alongside this data, we plotted the parental allelic proportion at the 15 false-negative Alu loci described above (Supplementary Fig. 2a). To formalize this result and ensure that these specific loci do not have higher genotyping or sequencing error than other loci, we performed three analyses. First, we compared the combined allelic proportion of all heterozygous variants to the combined allelic proportion of both false-negative parents and the associated heterozygous probands (Supplementary Fig. 2b). Second, we examined the allelic proportion for all homozygous reference genotypes at each false-negative locus for evidence of higher error rates based on higher average allelic proportion at those loci using a z-test (Supplementary Fig. 2c). Finally, to assess if these loci had similar read start distributions in both parent and proband, we retrieved all read start sites within 250 bps of each insert site and used a Kolmogorov–Smirnov test to compare the two distributions. Only one locus, chr11:g.95523918_95523919insAlu (1_280), showed a significant difference at a threshold of $p = 0.05$ (K.S. test $p = 2.70 \times 10^{-4}$).

Since these sites are not likely to be genotyping or sequencing artifact, we next attempted to clarify the nature of a subset of variants with a custom long-range PCR and Pacific Biosciences sequencing approach. The goal of this methodology

was to attempt to both validate and phase candidate mosaic MEIs to nearby (±5 Kbps) SNVs. As above, we limited our analysis to *Alu* MEIs as the large insert length of L1 ($n = 1$) and repeat heavy SVA ($n = 2$) elements would likely prove refractory to long PCR with enough flanking sequence to phase to nearby SNVs. To design locus specific primers, the genomic region surrounding each target MEI was entered into Primer-BLAST [www.ncbi.nlm.nih.gov/tools/primer-blast/] using default parameters. Each locus was targeted with three primer pairs: one pair amplifying a 10 kb region with the MEI lying centrally, and two pairs of primers to amplify 5 kb fragments extending up and downstream of the MEI. In total, 10 kb PCRs were performed using AccuTaq LA DNA Polymerase (D8045, Sigma-Aldrich) with the following parameters: 98° denaturation for 1 min, followed by 30 cycles of 94° (30 s)/54° (30 s)/68 degrees (11 min), with a final extension of 18 min at 68°. Total 5 kb PCRs were performed as above, but with a reduced 6 min extension time at 68°s, and a final extension time of 10 min. After quality checking the PCRs on control DNA, 100–200 ng of both parental and proband DNA were amplified in a 50 µl PCR reaction using the same parameters.

After confirming amplification by agarose gel, PCR products were isolated using SPRI bead purification (Ampure-XP, Agilent), quantified, and mixed in equimolar amounts. We then performed a standard PacBio library prep using SMRTbell Template Prep Kit 1.0 (part no. 100-991-900) without size selection. Sequencing was performed on the Pacific Biosciences Sequel platform using Sequel Sequencing Kit 3.0 (101-597-800) and Sequel SMRT cell 1 M v3 (101-531-000). For error correction, raw sequencing reads were then provided as input to pbccs v3.4.1 [https://github.com/PacificBiosciences/unanimity] with default parameters other than minLength = 100. Corrected reads were then aligned to the human genome using blasr v5.3.3[49] and downsampled to between 2000–3000× coverage per locus assessed.

MEIs were then phased to the nearest heterozygous SNV on a per-read basis using a custom java pipeline. After dropping sites due to PCR errors or phasing ambiguity, we were able to examine a total of 6 *Alu* MEIs (Supplementary Data 2). The total number of reads supporting each of four possible phases (MEI + SNV +, MEI + SNV−, MEI-SNV +, MEI-SNV−) were then quantified. Primers and quantification of the results are available in Supplementary Data 2.

**Quality control of RT data using WGS and 1KGP**. To determine if our MEI WES call set was biased compared with WGS data, we performed two independent comparisons: (1) to high coverage (>30 ×) WGS data generated for a subset of DDD trios; and (2) to a published collection of MEIs from 1KGP phase III[5].

For WGS quality control, we used a subset of 30 DDD trios ($n = 90$ individuals), which were previously whole-genome sequenced. MEI discovery using MELT[5] on all 90 individuals was performed and filtered identically to the WES data. Genotypes identified in the WGS data but not in WES were then separated based on coverage in the corresponding WES. Genotypes in low coverage areas (< 10 ×) were considered not possible (n.p), while variants where coverage was greater than 10x are considered not detected (n.d). All remaining genotypes were than compared for identity between the WGS and WES results (Supplementary Table 1).

To compare the DDD MEI call set to the 1KGP, we first filtered 1KGP calls to variants with mean > 10× coverage in 1000 randomly sampled DDD WES individuals (leaving 318 *Alu*, 81 L1, and 26 SVA). We then randomly selected 2453 DDD parents 1000 times, retaining only loci present in downsampled individuals[4,5]. The resulting distribution was then compared with the observed number of variants in the 1KGP-masked data to generate z-scores independently for all three MEI types (Supplementary Fig. 4).

To compare our PPG data set to Zhang et al.[6], we downloaded provided Supplementary tables. We then summed the total number of unique events per person, and determined counts for each gene reported. Genes were then matched between our call set and Zhang et al.[6] using ENSEMBL gene identifiers, and allele counts between each data set were plotted to create Supplementary Fig. 5.

**GTEx annotation of processed pseudogenes**. To determine RNA expression levels of donor genes which gave rise to PPGs identified in this study, we queried transcript per kilobase per megabase of sequencing (TPM) scores for all genes in 30 tissues assessed by the current GTEx v7 release [https://gtexportal.org/home/datasets]. Only the 18,225 protein-coding genes which were assessed for gene PPGs by our project were retained for subsequent analysis. TPM values were then averaged across all GTEx individuals for a given tissue to generate a mean TPM value as plotted in Supplementary Fig. 6. Nonparametric Wilcoxon rank-sum tests were performed using the wilcox.test function in R with default parameters to generate *p*-values for both within tissue and between tissue comparisons.

**SNV calling and quality control**. To call SNVs from all DDD individuals, we utilized GATK 3.5-0-ge91472d[50] in three steps using default settings. First, we called variants in individual samples using HaplotypeCaller. Next, individual VCF files were processed in 200 individual batches using CombineGVCFs. Finally, all batched VCFs were passed to GenotypeGVCF to generate a final joint-called VCF file. This file was then annotated using VEP v88[44] on build GRCh37 of the human genome; final annotations were determined only for the canonical transcript as determined by VEP. Unaffected parents ($n = 17,032$ individuals) were then

extracted from this VCF and only variants with an allele count ≥ 1 in these individuals were retained.

For initial filtering, we removed SNVs with a VQSLOD < −2.0, read depth < 8, and genotype quality < 20. We next performed more extensive QC using a "missingness" score identical to the method described in Martin et al.[13]. In short, each genotype at a given variant was assessed for genotype quality (GQ), depth (DP), and a binomial test for allelic depth (i.e., number of alternate vs. reference supporting reads; AD). If a given genotype had GQ < 20, DP < 8, or AD *p*-value < 0.001, it was considered "missing". If more than 50% of genotypes for a given variant were missing, the variant was subsequently filtered from final analysis. Allele frequencies were recalculated based on included individuals while accounting for missing genotypes.

**Selective constraint of SNVs and MEIs**. As sensitivity of variant discovery can bias our results, we generated an accessibility mask of the DDD WES data where we expect our variant ascertainment sensitivity to be > 95% (Supplementary Fig. 11)[5]. Our mask thus includes only regions of the genome that contain at least 10× average coverage in a mean cohort of 1000 randomly selected individuals for a total of 74.2 Mbp, or ~2.3% of the genome (Supplementary Table 3). Using this mask, we filtered our original 1129 variants down to 790 (653 *Alu*, 107 L1, and 30 SVA) variants in unaffected parents ($n = 17,032$ individuals). Parents were determined to be affected either by the referring clinician or, where ambiguous, through manual curation of HPO terms for a matching parent–offspring phenotype.

Using this masked subset of variants, we determined genomic constraint as shown in Fig. 2b. Allele frequency values were recalculated for all variants, and a pLI score[22] for each MEI was added as described above. MEIs which did not insert into a gene or inserted into a gene without a calculated pLI score[22] were excluded from subsequent analysis. We then calculated proportion of singleton variants and proportion of variants in high pLI genes independently for *Alu* and, due to low overall numbers of the other MEI subtypes, for a combined set of *Alu*, L1, and SVA. SNVs annotated as nonsense, missense, synonymous, or splice acceptor/donor (splice in Fig. 2b) as determined by VEP v88[44] were extracted from the SNV VCF files described above and used to calculate singleton and pLI proportion identically to MEIs.

**Fetal brain expression data**. To assess if genes disrupted by RT variants were expressed in the fetal brain, we downloaded per-gene expression data generated by the BrainSpan consortium (file: genes_matrix_csv.zip [http://www.brainspan.org/static/download.html])[30]. From this file, we extracted columns representing data generated from fetuses 15 to 37 weeks post conception. These data were then provided as input to a custom python script which calculated the median RPKM of all genes provided for the file. Genes with a median RPKM ≥ 1 were considered expressed in the fetal brain ($n = 15,474$). Likely causal RT variants ($n = 4$) were then intersected with this list to determine gene expression status. As RPKM ≥ 1 is a rather liberal cutoff, we also ranked these genes compared with other genes in our expression list; these values are reported in Supplementary Table 4.

**MEI validation by PCR**. To validate all nine de novo MEI variants (Table 2) and the homozygous insertion in *PAN2*, we used the following PCR protocol: primers were designed using Primer3[51] to make products spanning the predicted insertion site (Supplementary Data 1). PCR was carried out using Platinum™ Taq DNA Polymerase High Fidelity (Invitrogen); 20 ng of genomic DNA extracted from blood or saliva was amplified in the presence of 0.2 µM of each primer and 1 unit of Platinum™ Taq. Amplification was carried out using the following cycling conditions; for Alu insertions: 2 min at 94 °C, followed by 36 cycles of (30 s at 94 °C, 30 s at 60 °C, and 1 min at 68 °C); for LINE1 insertions: 2 min at 94 °C, followed by 36 cycles of (30 s at 94 °C, 30 s at 60 °C, and 7 min at 68 °C). PCR products were visualized using a 2% agarose E-Gel® (Invitrogen).

**Processed pseudogene validation**. To validate the two de novo PPG variants (Table 2), we used the following PCR protocol: primers were designed using Primer3[51] to make products within the exons of each gene. Forward and reverse primers were then paired between exons to amplify across the excised intronic regions (Supplementary Data 1). PCR was carried out using either Platinum™ Taq DNA Polymerase High Fidelity (Invitrogen) or Thermo-Start Taq DNA Polymerase (Thermo Scientific). Platinum™ Taq assay: 20 ng of genomic DNA extracted from blood or saliva was amplified in the presence of 0.2 µM of each primer, and 1 unit of Platinum™ Taq. Amplification was carried out using the following cycling conditions; 2 min at 94 °C, followed by 36 cycles of (30 s at 94 °C, 30 s at 60 °C, and 1 min at 68 °C). Thermo-Start Taq DNA Polymerase assay: 40 ng genomic DNA was amplified in the presence of 0.2 µM of each primer and 0.42 units of Thermo-Start Taq. Cycling conditions were as follows: 5 min at 95 °C, six cycles of (30 s at 95 °C, 30 s at 64 °C, and 1 min at 72 °C), six cycles of (30 s at 95 °C, 30 s at 62 °C, and 1 min at 72 °C), six cycles of (30 s at 95 °C, 30 s at 60 °C, and 1 min at 72 °C) followed by 36 cycles of (30 s at 95 °C, 30 s at 58 °C, and 1 min at 72 °C) with a final elongation of 10 min at 72 °C. PCR products were visualized using a 2% agarose E-Gel® (Invitrogen). PCR products were sequenced using either the forward or reverse primer used in the amplification protocol by Eurofins GATC Biotech

GmbH. Sequence traces were aligned using SeqMan Pro 15 (Lasergene 15) and reads were aligned to the human genome (build GRCh37) using BLAT[52].

**WGS of probands with de novo processed pseudogenes**. To validate and determine the insertion site of the two identified de novo PPGs (Table 2), we performed Illumina WGS on all individuals of each trio in which the de novo event was identified ($n = 6$ individuals). Samples were first quantified with Biotium Accuclear Ultra high sensitivity dsDNA Quantitative kit using Mosquito LV liquid platform, Bravo WS and BMG FLUOstar Omega plate reader and cherrypicked to 500 ng/120 μl using Tecan liquid handling platform. Cherrypicked plates are then sheared to 450 bp using a Covaris LE220 instrument and subsequently purified using SPRI Select beads on Agilent Bravo WS. Library construction (ER, A-tailing and ligation) was performed using "NEB Ultra II custom kit" on an Agilent Bravo WS automation system. Samples were then tagged using NextFLEX Unique Dual Indexed adapter 1–96 barcodes at the ligation stage. Libraries were then quantified by qPCR using Kapa Illumina ABI Sanger custom qPCR kits using a Mosquito LV liquid handling platform, Bravo WS, and Roche Lightcycler. Libraries are then pooled in equimolar amounts on a Beckman BioMek NX-8 liquid handling platform and normalized to 2.4 nM for cluster generation on a c-BOT and then sequenced on the Illumina TenX sequencing platform. Following sequencing, reads were aligned with BWA mem[42] (with settings -t 16 -p -Y -K 100000000) to build GRCh37 of the human reference genome. Reads were then manually inspected using the Integrative Genomics Viewer (IGV)[53] to confirm presence, de novo status, and parent of origin of each PPG.

**Methylation analysis**. Methylation analyses were performed in accordance with the methods detailed in Choufani et al.[32]. In short, we downloaded publicly available methylation data from GEO (accession GSE74432) generated by Choufani et al.[32] and combined it with Illumina EPIC methylation array data generated in-house for our *Alu NSD1* loss-of-function patient (Fig. 3b; Table 2). Following data curation, we randomly sampled a subset of 15 cases (defined by Choufani et al.[32] as a carrier of an *NSD1* loss of function variant) and 15 controls to identify a set of differentially methylated signature CpGs based on a 20% effect size cutoff and 0.01 FDR-corrected $p$-value which separated case and control individuals. Once a signature was established, we used Pearson correlation to assess each sample's similarity to a mean case and mean control based on differential methylation at signature CpG sites. Confirmed pathogenic *NSD1* variants and control samples used to build the signature were not included in the correlation analysis (Supplementary Fig. 10).

**MEI mutation rate and burden**. To determine the mutation rate independently for each MEI type (*Alu*, L1, SVA), we utilized data generated by both DDD and the 1KGP[5]. For DDD data, we filtered sites as above based on our > 10× coverage accessibility mask. For the 1KGP data[5], we created a combined mask from three different data sources: (1) the pilot accessibility mask generated by the 1KGP project phase III[54], which removes regions of the genome inaccessible to variant calling, (2) reference ME sequences as identified by repeatmasker[55], as MELT is unable to accurately ascertain MEIs in these regions, and (3) All sequence ± 10 Kbp from the 5′ and 3′ terminus of all protein-coding genes from RefSeq[56]. This mask was generated separately for *Alu* and L1 and did not filter 1113.0 Mbp or 959.9 Mbp of the genome, respectively. The *Alu* mask was used for filtering SVA and both masks excluded both allosomes. On masking the 1KGP data, we were left with a total of 10,930 autosomal MEIs (8,554 *Alu*, 2,047 L1, 329 SVA). Following filtering of the DDD and 1KGP sets with their corresponding masks, we used the Watterson estimator with an effective population size of 10,000 for all calculations to estimate the population mutation parameter, $\Theta$[33], and mutation rate, $\mu$ (Supplementary Table 3).

We next used our estimate of $\mu$ to determine the expected number of de novo MEI mutations in exons, enhancers, and introns genome-wide. To calculate this value, we extrapolated $\mu$ for 9738 births (i.e., number of probands in our study), resulting in an estimate of 677 de novo MEIs genome wide. We then simulated this number of variants 100 times genome-wide using bedtools random[45] and annotated the resulting simulated variants identically to real variants reported in this study (e.g., as in Fig. 2a). The total number of expected variants in the three categories depicted in Fig. 4 were then averaged across all simulations to determine Poisson $\lambda$ of de novo variants under neutral mutation rate. These distributions were then statistically compared with the number of observed variants in DDD using the *ppois* function in R (Fig. 4).

**Reporting summary**. Further information on research design is available in the Nature Research Reporting Summary linked to this article.

## Data availability

Sequencing, phenotype data, and variant calls for all data in this paper are accessible via the European Genome-phenome Archive (EGA) under study number EGAS00001000775 [https://www.ebi.ac.uk/ega/studies/EGAS00001000775]. Within this study, WES files of all DDD families are provided as part of data set EGAD00001004390,

and RT variant calls and WGS data of the two individuals with de novo PPG variants are provided as part of data set EGAD00001004586.

## Code availability

We have deposited R code, source data files, and raw blots from PCRs used to generate main text and Supplementary figs. online at the following github repository [https://github.com/eugenegardner/MEIPaper.git]. Source code and more information for the PPG discovery pipeline used in this paper is available online at github [https://github.com/eugenegardner/Retrogene.git].

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

## Acknowledgements

The authors wish to thank the Wellcome Sanger Institute sequencing facility staff for their assistance in preparing samples and performing sequencing experiments, all members of the DDD study for providing valuable comments during data analysis and the paper preparation, and the DDD families—this work would not be possible without their confidence and support. We also thank Panayiotis Constantinou for helping to curate known MEI-associated cases and for annotation of affected parents as well as Hilary Martin for constructive comments during manuscript preparation. We also wish to acknowledge Jeffrey Barrett and Caroline Wright for their leadership of the DDD. The DDD study presents independent research commissioned by the Health Innovation Challenge Fund [grant number: HICF-1009-003], a parallel funding partnership between Wellcome and the Department of Health, and the Wellcome Sanger Institute [grant number: WT098051]. The views expressed in this publication are those of the author(s), and not necessarily those of Wellcome or the Department of Health. The study has UK Research Ethics Committee approval (10/H0305/83, granted by the Cambridge South REC, and GEN/284/12 granted by the Republic of Ireland REC). The research team acknowledges the support of the National Institute for Health Research, through the Comprehensive Clinical Research Network. This study makes use of DECIPHER [http://decipher.sanger.ac.uk]), which is funded by the Wellcome.

## Author contributions

E.J.G. performed RT variant calling and annotation, PPG algorithm design, constraint and burden testing, computational analysis of mosaic variants, and initial clinical annotation and together with M.E.H. designed experiments, oversaw the study, and wrote the paper. E. P. designed and performed PCR experiments. G.G. curated and prepared DDD sequencing data. S.S.G. and H.I. designed and conducted molecular assays to assess mosaic MEIs. P.J.S. assisted in estimating genetic burden of deleterious MEIs in the human population. A.S. assisted with the design of the PPG discovery algorithm. P.D., K.E.S., E.J.G., and T.S. performed variant calling and quality control of SNVs. J.H. performed methylation analysis of the *NSD1* variant patient. K.E.C., E.C., K.L. L., K.P., E.R., D.R.F., and H.V.F. prepared clinical assessments of patients and confirmation of molecular diagnoses as they relate to patient phenotype.

## Competing interests

M.E.H. is a co-founder of, consultant to, and holds shares in, Congenica Ltd, a genetics diagnostic company. The remaining authors declare no competing interests.
