## [Peer Review File · Nature Communications]

Reviewers' comments:

Reviewer #1 (Remarks to the Author):

This study is the latest installment from DDD. Here, they move to the evaluation of MEIs, a potential class of deleterious coding variation that has not been previously assessed. They applied MELT for MEIs, and a method developed for detection of pseudogenes from WES (for which they have already provided source code in github, a readme, and an update for publication). The overall study included WES from 9,738 DDD study trios and another 90 WGS trios from the cohort for comparisons. Both methods have been carefully benchmarked against available data and previous studies, and the comparisons all support this being a comparable call set to what might be generated from WGS in coding regions. The results are analyzed for several obvious questions – de novo variation, causal variation, mutation rate estimates, and overall selection against MEIs. The study is appropriately conservative, as are the conclusions, which estimate a causal coding MEI in ~0.04% of cases, and go as far as to estimate the plausible contribution from WGS. The paper is well written and clear on all analyses. These data should be published and are important benchmarks for the field. I have only minor comments that might be of use:

1. Mention the 90 WGS samples in the comparison of WES vs WGS – I had to go to methods to find this (lines 102-104)

2. Helpful to provide a breakdown of the false findings with counts of false positive in proband vs false negative in parents? Any evidence of germline mosaicism in the latter group? You could adapt the pipeline to re-genotype the parents in the process of all de novo predictions to clean up the false negatives during analyses – though the events are so rare and easily reviewed it may be irrelevant (the note of the 15 minute review process was amusing).

3. The EZR variant is interesting. Agreed it cannot be deemed diagnostic, but is notable and warrants reassessment in the future. SETD5 – was there any evidence of a compound het from additional coding variants? Also, did you cross reference the insertion site localization with brain-expressed transcripts for the few 'causal' variants – e.g. MEF2C?

4. Line 106-108 - Were there any signature of MEIs only identifiable with WGS, or was this simply a matter of capture efficiency or proximity to the edges of the capture regions?

Reviewer #2 (Remarks to the Author):

The manuscript by Gardner et al. reports on the discovery and analysis of retrotransposition-derived variants, mobile element insertions (MEIs) and processed pseudo genes (PPGs), in 9,738 whole-exome sequenced (WES) probands with developmental disorders (DDD study) and their 18,394 WES parents. The manuscript provides interesting insights into the contribution of retrotransposition. Only few large-scale but smaller studies of these types of structural variants have been published to date and the analysis of MEIs and PPGs are not yet part of routine clinical sequencing pipelines. Here, a new tool for the identification of PPGs was developed implementing a straight-forward approach and an existing tool, MELT, was used for the discovery of MEIs. The main biomedical findings reported by the authors include new estimates of mean retrotransposition sites in the human exome per individual, estimates of de novo MEI rates, a selective constraint that is similar to nonsense and splice-site SNVs for exonic retrotransposition events, and the identification of four likely pathogenic retrotransposition mutations. As stated by the authors in the discussion, the overall burden of disease attributable to retrotransposition is low: 0.04% of diagnoses in the DDD study. However, four pathogenic events were identified and I agree with the

authors that it is, nevertheless, an important class of variants to assess when considering individual patients.

Specific comments:

1) Lines 412-413: For the new tool, "Source code and more information is available online at github [link]". The "more information" is missing on github, no documentation is provided. Please make clear whether the tool was developed to specifically analyze the DDD cohort only or whether it can be applied to other cohorts, too. In the latter case, please provide documentation, i.e. at least install and running instructions.

2) Lines 117-119: How did you ascertain that the number of variants approximated a Poisson distribution?

3) Lines 165-167: The statements that you assayed only 2.2% of the genome and that you ascertained over five-fold more variants in exons should be split into separate sentences. The current sentence is misleading implying that even though you assayed only a small fraction of the genome, you identified five times more variants in the exome. However, you assayed the entire exome which amounts to 2.2% of the genome.

4) Line 187: Please write out protein-truncating variant before first using the abbreviation (PTV).

5) Lines 187, 295, and 315-316: I am confused how MEIs can make up roughly 1% of all coding PTVs, but only 0.04% of DD cases is directly attributable to RT-derived mutagenesis even though there is a significant enrichment of de novo MEIs in dominant DD genes. I understand that not all PTVs have an effect on disease and that not all DD cases can be explained by mutations in dominant DD genes. Am I mistaken by assuming that 1% of all coding PTVs in dominant DD genes should be MEIs and thus candidates for pathogenic variants? Is this discrepancy due to selection against PTVs in dominant DD genes?

6) Lines 318-320: Please explain why you "assessed the probability that four out of *six* exonic de novo MEIs would fall within exons of dominant DD-associated genes by chance".

7) Lines 37-38 and 97: It is stated that the manuscript describes the *single* largest study of this kind while the discussion mentions that it is the largest study alongside Torene et al. (line 326). Please correct.

8) Line 469: Should be SNV calling, not SNV variant calling as the V in SNV stands for variant.

9) Lines 469-474: What was the reason for re-running the SNV calling (in batches of 200 individuals) instead of using the calls from the original DDD study (ref. 12)?

10) Line 487: I was initially misled by the heading, interpreting it that you *constrain* the genome to accessible regions instead of understanding that you refer to the "selective constraint". I'd suggest to rename the section to "Selective constraint of SNVs and MEIs".

11) The manuscript is well written and the language is clear. Only the last paragraph of the methods section is somewhat written in note form with missing articles.

12) Most statements are supported by statistical tests. I am not a statistician but to me the analyses look statistically sound.

Reviewers' comments:

Reviewer #1 (Remarks to the Author):

This study is the latest installment from DDD. Here, they move to the evaluation of MEIs, a potential class of deleterious coding variation that has not been previously assessed. They applied MELT for MEIs, and a method developed for detection of pseudogenes from WES (for which they have already provided source code in github, a readme, and an update for publication). The overall study included WES from 9,738 DDD study trios and another 90 WGS trios from the cohort for comparisons. Both methods have been carefully benchmarked against available data and previous studies, and the comparisons all support this being a comparable call set to what might be generated from WGS in coding regions. The results are analyzed for several obvious questions – de novo variation, causal variation, mutation rate estimates, and overall selection against MEIs. The study is appropriately conservative, as are the conclusions, which estimate a causal coding MEI in ~0.04% of cases, and go as far as to estimate the plausible contribution from WGS. The paper is well written and clear on all analyses. These data should be published and are important benchmarks for the field. I have only minor comments that might be of use:

1. Mention the 90 WGS samples in the comparison of WES vs WGS – I had to go to methods to find this (lines 102-104)

We have changed the wording of this sentence to: “We compared MEI variants identified by MELT within the DDD WES data to both WGS data generated for 90 overlapping DDD individuals and population MEI data previously generated from the 1000 Genomes Project Phase 3 (1KPG)⁴⁻⁶ WGS data.”

2. Helpful to provide a breakdown of the false findings with counts of false positive in proband vs false negative in parents? Any evidence of germline mosaicism in the latter group? You could adapt the pipeline to re-genotype the parents in the process of all de novo predictions to clean up the false negatives during analyses – though the events are so rare and easily reviewed it may be irrelevant (the note of the 15 minute review process was amusing).

We have added the breakdown of FP sites vs FN genotypes to the text: “(1 incorrect candidate *de novo* variant per every 295 patients; either a false positive variant [1 per 649] or false negative genotype [1 per 541] in at least one parent).”

To address the question of parental germ-line mosaicism, we re-analysed the parents, similar to as suggested by the reviewer, by quantifying reference and alternate read support through “re-genotyping” all *Alu* loci reported in this study in 1,000 randomly selected trios and the 15 trios in which we identified a potentially mosaic *Alu* variant. This allowed us to quantify the proportion of all reads supporting the insertion at these sites (allelic proportion), and thus test whether the parents exhibited signs of mosaicism. Supplementary Figure 2 shows this result. As evidenced by this plot, potential parental mosaic variants (as grey bars) typically exhibit a low allelic proportion overlapping the distributions observed for all homozygous reference and heterozygous genotypes. As shown by Supplementary Figure 2c, when we take a per-locus approach, 12 out of 15 putative mosaic variant's allelic proportions are clearly and statistically different from homozygous reference genotypes at those same loci. This also suggests that overall error rates at these loci are low; however, based on this approach, it is difficult for us to determine whether these sites are truly mosaic or if they represent a class of variant which MELT overall struggles to genotype due to low alternate read support. Nonetheless, they do strongly support the contention that these calls are not false positives in the offspring.

We also attempted to phase *Alu*, L1, and SVA potential parental mosaic variants using long-range PCR and PacBio sequencing encompassing surrounding SNVs – similar to the approach we used to phase *de novo* variants and as reported in main text table 2. We were able to use this method for two out of 15 such variants, both *Alu* insertions, due to the lack of heterozygous SNVs within the fragment length of the WES read library. One of these variants, an *Alu* at position 19:40407986, does appear to be mosaic, with 4/11 (36.4%) of reads that contain a heterozygous SNV at position g.19:40408128T>C that cross the insertion breakpoint containing evidence supporting the MEI. At the

other *Alu* variant, phasing with a nearby heterozygous SNV showed no evidence of mosaicism. In light of this analysis, we have added the following to the main text:

“We investigated whether false negative parental genotypes could be mosaic. Using a long-PCR coupled to long-read sequencing assay (see methods), we were able to phase four of our false-negative *Alu* variants to nearby SNVs. With our approach, we could demonstrate two insertions were mosaic in a parent (chr3:g.101378585_101378586ins*Alu*(1_280), chr15:g.91145646_91145647ins*Alu*(1_281)), one was part of a complex event of unknown mosaicism (*FCGBP*:c.4743_4744ins*Alu*(1_110)), and another constitutive. Therefore these false apparent *de novo* calls can be generated by a variety of causes, including parental mosaicism. (Supplemental Fig. 2).”

3. The EZR variant is interesting. Agreed it cannot be deemed diagnostic, but is notable and warrants reassessment in the future. SETD5 – was there any evidence of a compound het from additional coding variants? Also, did you cross reference the insertion site localization with brain-expressed transcripts for the few ‘causal’ variants – e.g. MEF2C?

We agree that the EZR variant deserves reassessment in the future, we are looking out for similarly damaging variants in this gene in other datasets.

Regarding the patient (Decipher ID 280818) with the deleterious SETD5 *Alu* insertion, we considered three possible scenarios likely to relate to the patient’s unusual phenotype for this disorder: 1.) Additional potentially causative *de novo* variants of other classes, 2.) Additional variants in SETD5 and 3.) Inherited compound heterozygous variation in known recessive genes. Additional scenarios (i.e. inherited variants not in compound heterozygosity) are difficult to interpret. The patient has 4 additional *de novo* variants, all of which are either non-coding, located in genes which have no prior association to DD, nor have any indication of being LoF intolerant (i.e. high pLI scores). Secondly, the patient also has neither a *de novo* nor an additional protein-modifying mutation in SETD5. Finally, we also did not identify any inherited compound heterozygous variation in known recessive genes. To clarify this patient’s findings, we have added the following sentence to the main text:

“Accordingly, we did investigate this patient’s exome for other potentially diagnostic variants, but did not identify additional *de novo* or inherited variation that could plausibly be associated with this patient’s phenotype.”

To assess the brain expression of the four genes with apparently diagnostic MEIs, we downloaded fetal brain gene expression data from BrainSpan and selected prenatal samples (15 to 37 weeks post-conception) and calculated median reads per kilobase per megabase (RPKM) values per gene using all samples. We then filtered out genes from this list with RPKM < 1 to broadly select genes expressed in the fetal brain. We then interrogated the disrupted gene for each of our likely causative variants (main text Fig 3a-d). All genes were expressed by this metric, with specific RPKM values of (percentile in parenthesis):

SETD5 – 10.00 (65.3%ile)
NSD1 – 9.26 (63.2%tile)
ARID2 – 4.36 (38.5%tile)
MEF2C – 84.37 (96.7%tile)

We have modified the following text to show this finding:

“Of the mutations identified, four directly inserted into coding exons of DD-associated and fetal brain-expressed (see methods; Supplemental Table 5)³³ genes (Fig. 3, Table 2) with all four found in genes statistically enriched for PTVs¹² and therefore likely to operate by a LoF mechanism.”

We have added the section “Fetal Brain Expression Data” to the methods to describe how this analysis was performed.

4. Line 106-108 - Were there any signature of MEIs only identifiable with WGS, or was this simply a matter of capture efficiency or proximity to the edges of the capture regions?

Our conclusion is indeed that MELT loses sensitivity as coverage drops towards the edge of WES bait capture regions. This conclusion is supported by our findings depicted in Supplemental Fig. 11 whereby if we limit to regions with >10X coverage, our allele frequency distribution (for *Alu* and L1) more closely approximates that of SNVs. This finding is why our more sensitive analyses depicted in main text Figs. 2b and 4 use sites that were in regions where mean coverage across all individuals is >10X.

Reviewer #2 (Remarks to the Author):

The manuscript by Gardner et al. reports on the discovery and analysis of retrotransposition-derived variants, mobile element insertions (MEIs) and processed pseudo genes (PPGs), in 9,738 whole-exome sequenced (WES) probands with developmental disorders (DDD study) and their 18,394 WES parents. The manuscript provides interesting insights into the contribution of retrotransposition. Only few large-scale but smaller studies of these types of structural variants have been published to date and the analysis of MEIs and PPGs are not yet part of routine clinical sequencing pipelines. Here, a new tool for the identification of PPGs was developed implementing a straight-forward approach and an existing tool, MELT, was used for the discovery of MEIs. The main biomedical findings reported by the authors include new estimates of mean retrotransposition sites in the human exome per individual, estimates of de novo MEI rates, a selective constraint that is similar to nonsense and splice-site SNVs for exonic retrotransposition events, and the identification of four likely pathogenic retrotransposition mutations. As stated by the authors in the discussion, the overall burden of disease attributable to retrotransposition is low: 0.04% of diagnoses in the DDD study. However, four pathogenic events were identified and I agree with the authors that it is, nevertheless, an important class of variants to assess when considering individual patients.

Specific comments:

1) Lines 412-413: For the new tool, "Source code and more information is available online at github [link]". The "more information" is missing on github, no documentation is provided. Please make clear whether the tool was developed to specifically analyze the DDD cohort only or whether it can be applied to other cohorts, too. In the latter case, please provide documentation, i.e. at least install and running instructions.

We have updated the github site with documentation to ensure others can use our tool on their own datasets.

2) Lines 117-119: How did you ascertain that the number of variants approximated a Poisson distribution?

To clarify this point, we have added the following to the methods in the section "MEI call set generation and consequence annotation":

"To determine if variants per individual approximated a Poisson distribution, we calculated the mean number of sites per unrelated individual (λ) independently for each RT class and for all RT classes combined as displayed in Fig. 1a-e. This value was then supplied to the function *rpois* in R to generate a random deviate at that λ and was plotted alongside histograms in Supplemental Fig. 3."

This addition is also noted in the main text: "As expected, the total number of variants per individual for each RT class (Fig. 1a-d) as well as combined number of RT events (Fig. 1e) approximated a Poisson distribution (see methods; Supplemental Fig. 3)."

While a formal statistical test comparing the observed and Poisson distributions does show statistically significant differences, the visual comparison shows that these are subtle differences, hence why we use the term "approximated". If the editor prefers we could be somewhat more explicit and say something along the lines of "... was similar to a Poisson distribution, with subtle but statistically significant differences, ..."

As noted above, we have added a supplemental figure that shows this result.

3) Lines 165-167: The statements that you assayed only 2.2% of the genome and that you

ascertained over five-fold more variants in exons should be split into separate sentences. The current sentence is misleading implying that even though you assayed only a small fraction of the genome, you identified five times more variants in the exome. However, you assayed the entire exome which amounts to 2.2% of the genome.

We have reworded this sentence, splitting into two separate sentences to improve clarity.

“Due to the large numbers of individuals with WES data in this study, we have ascertained over five-fold more exonic variants than the largest previously published study (Supplemental Fig. 7)^{4,5}. Nonetheless, the number of MEIs identified in this study, based on the proportion of the genome assayed, likely represent only 2.2% of MEI variants genome-wide in these individuals.”

4) Line 187: Please write out protein-truncating variant before first using the abbreviation (PTV).

We have corrected this. The text now reads: “...MEIs thus make up roughly 1% of all coding protein truncating variants (PTVs; among SNVs, InDels, and large CNVs) in each individual human genome.”

5) Lines 187, 295, and 315-316: I am confused how MEIs can make up roughly 1% of all coding PTVs, but only 0.04% of DD cases is directly attributable to RT-derived mutagenesis even though there is a significant enrichment of de novo MEIs in dominant DD genes. I understand that not all PTVs have an effect on disease and that not all DD cases can be explained by mutations in dominant DD genes. Am I mistaken by assuming that 1% of all coding PTVs in dominant DD genes should be MEIs and thus candidates for pathogenic variants? Is this discrepancy due to selection against PTVs in dominant DD genes?

The 1% calculation is presented as a proportion of all exonic PTVs per sample, across all genes, and is presented approximately. A more accurate calculation would be 0.6% (0.76/120: Lek et al identify 120 exonic PTVs per individual, the average number of exonic MEIs per individual in this study is 0.76). All things being equal, we agree with the reviewer that one might expect 0.6% of pathogenic de novo PTVs in known DD-associated genes to be MEIs. About 10% of our cohort have de novo PTVs in known DD-associated genes, thus one might expect that 0.06% of the cohort might have pathogenic de novo MEIs in the same set of genes. Our finding of 0.04% is a little under this expectation, but given the small numbers of counts of pathogenic MEIs (observed=4, expected=6), these findings are consistent. We also note that the estimate of an average of 0.76 exonic MEIs per individual has a wide standard error, due to the low number of counts per individual, and is dominated by a small number of common MEIs.

For the sake of precision we have changed the text from “MEIs thus make up approximately 1% of all protein truncating variants...” to “MEIs thus make up 0.6% of all coding protein truncating variants”.

6) Lines 318-320: Please explain why you “assessed the probability that four out of *six* exonic de novo MEIs would fall within exons of dominant DD-associated genes by chance”.

We performed this analysis because it is robust to mis-estimating the genome-wide MEI mutation rate, and only requires the assumption that MEI mutation rates are approximately uniform across different exonic and intronic sequences. This analysis provides additional confidence that the number de novo MEIs in DD-associated genes represents a statistically significant enrichment, and is not driven by mis-estimating the genome-wide MEI mutation rate. We performed the analysis in this way as we know our sensitivity to detect MEIs is best in the coding sequence (i.e. exons) of genes that we capture with our WES baits. To test this assumption, we instead changed the test to the probability that four out of nine overall *de novo* MEIs (thus including those in introns) fall within exons of dominant DD-associated genes, and base this proportion on our accessible genome region (7.4×10^7 bps). We receive a similar p-value to the original test based on estimating a genome-wide MEI mutation rate – $p = 9.2 \times 10^{-5}$.

7) Lines 37-38 and 97: It is stated that the manuscript describes the *single* largest study of this kind while the discussion mentions that it is the largest study alongside Torene et al. (line 326). Please correct.

We have removed this statement in the abstract and replaced it with:

“Overall, our analysis represents a comprehensive interrogation of the impact of retrotransposition on protein coding genes and a framework for future evolutionary and disease studies.”

We have also modified the line “As our study is the first to discover MEIs directly from WES on a large scale...” to “As our study is the first to discover MEIs directly from WES on a large scale with MELT...”

8) Line 469: Should be SNV calling, not SNV variant calling as the V in SNV stands for variant.

This oversight has been corrected, the section header is now: “SNV Calling and Quality Control”

9) Lines 469-474: What was the reason for re-running the SNV calling (in batches of 200 individuals) instead of using the calls from the original DDD study (ref. 12)?

As part of the review process, we have developed a new quality controlled SNV call set that we feel better reflects current norms in SNV quality control. This updated call set has been used for all SNV-related figures/text and the methods section “SNV Calling and Quality Control” has been updated to reflect this change – along with clarifying exactly what was done.

To answer your question “instead of using the calls from the original DDD study (ref. 12)?”, calls reported in ref. 12 are only *de novo* and do not reflect population-level variants. We understand the confusion with our statement in the methods “Sequencing and SNV/InDel calling of families were performed as previously described” and have changed it to “Sequencing of families was performed as previously described” and clarified the section “SNV calling and quality control.”

10) Line 487: I was initially misled by the heading, interpreting it that you *constrain* the genome to accessible regions instead of understanding that you refer to the "selective constraint". I'd suggest to rename the section to "Selective constraint of SNVs and MEIs".

Thank you for the suggestion and we have updated the section header accordingly.

11) The manuscript is well written and the language is clear. Only the last paragraph of the methods section is somewhat written in note form with missing articles.

Thank you for the positive feedback. We have edited the final paragraph of the methods to be more stylistically in-line with the rest of the text.

12) Most statements are supported by statistical tests. I am not a statistician but to me the analyses look statistically sound.

REVIEWERS' COMMENTS:

Reviewer #1 (Remarks to the Author):

I had only minor considerations and considered the paper publishable on first review, and the authors have improved the manuscript further based on these minor suggestions and those of the other reviewer. I have no changes prior to publication.

- Michael Talkowski & Xuefang Zhao

Reviewer #2 (Remarks to the Author):

The authors have addressed all my comments well and in my opinion the manuscript is ready to be published in Nature Communications. Although appropriately answered by the authors, three short replies from my side:

- 1) Thanks for the documentation of the tool. I was successful in installing and running it on a small test case.
- 4) I agree that the differences are subtle in visual inspection and need not be mentioned in the main text, but in my opinion it should be stated in the Methods or Supplement that there are significant differences according to a formal statistical test.
- 6) Thank you for the explanation. I previously missed to read from Table 2 that you identified six exonic de novo variants.